# Childhood Vaccine Attitude and Refusal among Turkish Parents

**DOI:** 10.3390/vaccines11081285

**Published:** 2023-07-26

**Authors:** Osman Kurt, Osman Küçükkelepçe, Erdoğan Öz, Hülya Doğan Tiryaki, Mehmet Emin Parlak

**Affiliations:** 1Adiyaman Provincial Health Directorate, 02100 Adıyaman, Turkey; osman.kurt2@saglik.gov.tr (O.K.); erdogan.oz@saglik.gov.tr (E.Ö.); 2Gaziantep Provincial Health Directorate, 27310 Gaziantep, Turkey; hulya.dogantiryaki@saglik.gov.tr; 3Adıyaman Training and Research Hospital, 02100 Adıyaman, Turkey; meparlak02@gmail.com

**Keywords:** vaccine hesitancy, vaccine attitude, childhood vaccination, vaccine refusal

## Abstract

We aimed to understand and resolve anti-vaccine attitudes by examining the factors associated with vaccine attitudes and exploring potential strategies to improve childhood vaccination rates. Between 2014 and 2021, a total of 628 families refused vaccination in Adiyaman. A total of 300 families accepted visits and were visited. During the visits, the families were administered a questionnaire to determine the reasons for vaccine rejection and their opinions on the matter. While providing general information about the vaccine, parents were encouraged to reconsider their decision, and at the end, parents completed the questionnaire. The questionnaire included sociodemographic questions, reasons for vaccine refusal, and a vaccine attitude scale. Among the participants in the study, 9.3% were convinced about the vaccine. The mean vaccine attitude scale score was calculated as 23.6 ± 2.5 (min = 15–max = 29). Significantly higher rates of persuasion were observed among fathers (17.3%) compared to mothers (7.7%) (*p* = 0.038). Participants who had received some vaccinations had a higher rate of persuasion (11.6%) compared to those who had not received any vaccinations (2.6%) (*p* = 0.02). Childhood vaccine refusal is a complex issue that has been the subject of numerous studies. Studies on this subject will increase awareness of vaccines.

## 1. Introduction

Despite the availability of safe and effective vaccines, vaccine refusal remains a significant public health concern. Parental vaccine hesitancy can be a barrier to routine childhood immunization and contribute to a greater risk of vaccine-preventable diseases [1]. Factors such as lower educational level, lower household income, and distrust of the medical profession have been associated with vaccine hesitancy [2]. The COVID-19 pandemic has further complicated the issue, with some households reporting increased vaccine hesitancy [3]. To address this issue, it is important to understand the factors contributing to vaccine hesitancy/refusal and explore potential strategies to improve childhood vaccination rates [4].

In Turkey, vaccine refusal has increased over the past decade. Previously, cases of vaccination refusal were very rare but have increased rapidly since 2015 after the victory of a lawsuit for “obtaining parental consent for vaccination”, as well as frequent arguments against vaccination in the media. The number of families who refused to vaccinate their children was 183 in 2011, which increased to 980 in 2013, 5400 in 2015, 12,000 in 2016, and 23,000 in 2018 [5].

Several studies have been conducted to investigate vaccine attitudes in Turkey, particularly in relation to the COVID-19 vaccine. These studies provide insights into the levels of vaccine hesitancy, beliefs about the origin of the virus, and factors influencing vaccine acceptance among different populations in Turkey. One study by Salali and Uysal aimed to examine the levels of COVID-19 vaccine hesitancy and its association with beliefs on the origin of the novel coronavirus in a cross-cultural study between the UK and Turkey. The study conducted an online survey in Turkey with 3936 participants and found that 31% of the participants were unsure about getting vaccinated with the COVID-19 vaccine [6]. Another study by Bendau et al. explored the predictors of vaccine acceptance, including anxiety, fear, and individual risk perception. The study found that individuals with higher risk perception and more anxiety showed significantly higher vaccine acceptance in Turkey [7]. Karagöz and Yalman conducted a study to evaluate the attitudes, hesitancy, and confidence of healthcare professionals towards COVID-19 vaccines in Turkey. The study used a hospital-based cross-sectional research design and surveyed health workers from three different training and research hospitals in Istanbul. The findings revealed that nearly two-thirds of the participants had a positive attitude towards potential COVID-19 vaccines, which was associated with higher vaccine confidence and less hesitancy [8]. A study by Kaplan et al. focused on healthcare professionals and their attitudes towards the COVID-19 vaccine. The study found that about 84.6% of healthcare professionals in Turkey declared their willingness to accept the COVID-19 vaccine whenever possible. Factors influencing vaccine acceptance among healthcare professionals included advanced age, male gender, working in a primary health care center, living with family, having a child, having a chronic disease, and a higher fear of COVID-19 [9]. Topçu et al. conducted a study with mothers who refused vaccination in two different hospitals [10].

Overall, these studies provide valuable insights into vaccine attitudes in Turkey, highlighting the levels of vaccine hesitancy, the association between attitudes and vaccine acceptance, and the factors influencing vaccine attitudes among different populations, including healthcare professionals. Understanding these attitudes and factors is crucial for designing effective vaccination campaigns and addressing vaccine hesitancy in Turkey. The present study aims to obtain information about vaccine attitudes and evaluate the persuasion rate after people with vaccination hesitations are informed about vaccine attitudes. The present study aimed to determine the vaccination attitude, refusal, and persuasion rates according to the patient’s statements and objectively and definitively by examining the records.

## 2. Materials and Methods

### 2.1. Study Design and Location

This descriptive study was conducted in the Adiyaman province, located in the Southeastern Anatolia region of Turkey.

### 2.2. Study Population

The study population consisted of parents who had refused vaccination for their children by filling out vaccine rejection forms in Adiyaman. Between 2014 and 2021, a total of 628 families refused vaccination. The aim was to reach all of these families without using a sampling method. According to Turkish Statistical Institute data, the total number of births in Adiyaman province between 2014 and 2021 was 99,637. According to the sampling method with a 95% confidence interval, a non-response rate of 20% (it was thought that the non-response rate would be high in vaccine refusal), and a prevalence of vaccine refusal of 50% (the vaccine refusal rate was accepted as such since the vaccine refusal rate is not known precisely in the literature and the refusal rate is gradually increasing), it was seen that at least 306 families should be reached. All families who refused childhood vaccinations and filled out a refusal form were included in the study. Families with vaccine refusals who were visited at home were also included. Those who filled out a vaccine refusal form and were later convinced by the family physician’s advice and those who did not accept the home visit when called by phone were excluded. Initially, 628 families with vaccine refusals were called by telephone, and 615 could be reached by telephone. We told the families we contacted that we wanted to visit their homes, and the homes of 300 families who agreed were visited. All of the families were visited in their homes, and an interview was conducted with the parent at home for 30–60 min, depending on the parent’s compliance (Figure 1).

### 2.3. Study Instrument

The primary dependent variable in the study is whether or not the respondent is convinced about vaccination and vaccination behavior. The independent variables were age, interviewee, educational status, occupation, number of children, presence of children under one year of age, vaccination behavior, vaccination attitude, and reasons for not vaccinating. The visitors included general practitioners, nurses working in the directorate, and public health specialists among the authors.

Visits were made between 1 September 2022 and 31 March 2023. During the visits, the families were administered a questionnaire to determine the reasons for vaccine rejection and their perceptions on the matter. While providing general information about the vaccine, parents were encouraged to reconsider their decision, and at the end, parents completed the questionnaire. The questionnaire included sociodemographic questions, reasons for vaccine refusal, and a vaccine attitude scale. Families who were convinced to receive vaccinations were promptly supported in getting their vaccinations. Six months after the initial application, these families were contacted by phone to determine that they had vaccinated their children. In order to determine whether they were convinced about the vaccination, not only were their statements taken as a reference, but it was also determined through the information processing system that these families had been vaccinated.

The vaccine attitude scale (VAS), developed by Wallace et al., was used in this study. The scale’s Turkish validity and reliability study was conducted by Ceylan et al. [11,12]. It consists of 11 items on a 3-point Likert-type scale that measures parents’ attitudes towards vaccines. The scale covers four domains: “benefits of the vaccine” (1 and 2 items), “past vaccination behavior” (3 and 4 items), “efficacy and safety” (5 and 6 items), and “awareness of vaccine-preventable diseases” (7 and 8 items). It also includes a “confidence” sub-dimension (items 9, 10, and 11). Each item on the vaccine attitude scale is scored as agree = 1, undecided = 2, and disagree = 3. The total score is calculated by summing the individual item scores. The total score ranges from 11 to 33 points, where 11 indicates a positive attitude towards vaccination and 33 indicates a negative attitude. The scale does not have a specific cutoff point. Parents’ hesitancy towards vaccination increases as the score increases, indicating a negative attitude [12].

### 2.4. Ethics

The study obtained ethics committee approval from the Non-Interventional Research Ethics Committee of Firat University, with a decision dated 1 September 2022, and numbered 22 October 2022. Institutional permission was also obtained from the Adiyaman Provincial Health Directorate to conduct the research. Written and verbal consent was obtained from all participants, and the study followed the principles outlined in the Declaration of Helsinki.

### 2.5. Statistical Analysis

The statistical analysis was performed using the SPSS (Statistical Package for Social Sciences) software, version 22 (SPSS Inc., Chicago, IL, USA). Descriptive data were presented as frequencies and percentages for categorical variables and mean ± standard deviation (Mean ± SD) for continuous variables. The chi-square analysis (Pearson’s chi-square) was used to compare categorical variables between groups. The Kolmogorov–Smirnov test was employed to assess the normal distribution of continuous variables. The Mann–Whitney U-test was used to compare paired groups, while the Kruskal–Wallis test was used for comparisons involving more than two variables. The Spearman correlation test was conducted to examine the relationship between continuous variables. Statistical significance was set at *p* < 0.05.

## 3. Results

The study results showed that 300 participants, with a mean age of 34.2 ± 5.8 years (range: 21–53), were included in the analysis. Interviews were conducted with fathers in 17.3% of the cases and mothers in 82.7%. Regarding the participants’ education level, 8% had graduated primary/secondary school, 51.3% had graduated high school, and 40.7% had a university degree. In terms of occupation, 60% of the participants were homemakers. Regarding the number of children, 14% had one child, 35.7% had two children, and 50.3% had three or more children. Among the participants, 17% had children under the age of one.

Regarding the participants’ vaccination attitudes, 12.3% considered vaccines necessary, and 88% had vaccinated at least one of their children previously. Furthermore, 41.7% reported experiencing a problem related to vaccines. Regarding compliance with the Ministry’s recommended vaccines, 74.7% of the participants stated that their children had received some vaccinations, while 25.3% reported having not received any of the recommended vaccines (Table 1).

When participants were asked about the potential harms of vaccinating their children, the following responses were obtained: 39% believed that their children would “get sick more quickly”, 32% thought they would “contract deadly infectious diseases”, 29.3% expressed concerns about their children “getting sick often”, 23% believed their children would experience “a more severe prognosis of illness”, 14% stated that their children might become disabled after contracting a contagious disease, and 20.3% expressed concerns about other potential damages (Figure 2).

Among the participants included in the study, 89% stated that vaccines had side effects, 6% stated that they did not, and 5% were unsure. The most commonly reported side effects were pain (63.7%), fever (58.4%), infection (53.9%), rash (53.6%), infertility (26.6%), and other side effects (21.7%). Additionally, 1.5% of the participants were unsure if vaccines had any effects (Figure 3).

Among the study participants, 29.7% identified themselves as midwives/nurses/doctors; 76.3% were neighbors/friends; and 79.7% mentioned the media/press as their sources of information.

When asked about the reasons for not getting vaccinated, 99% said they did not believe the vaccines were safe or had concerns about their side effects. Additionally, 71.7% believed that vaccination was not necessary, 61% expressed doubts about the effectiveness of vaccines in protecting against diseases, 50.3% were concerned about the harmful chemicals in vaccines, 44.3% mentioned hearing about a child experiencing an adverse reaction after vaccination from another person, 36.7% attributed their decision to negative information they read or heard in the media about vaccines, 23.7% had negative experiences with healthcare personnel or institutions related to vaccination, and 8% cited religious reasons for not getting vaccinated (Table 2).

Among the participants in the study, 9.3% were convinced about receiving vaccines for their children. The mean vaccine attitude scale score was calculated as 23.6 ± 2.5 (min = 15 − max = 29).

Significantly higher rates of persuasion were observed among fathers (17.3%) compared to mothers (7.7%) (*p* = 0.038). Moreover, fathers had a significantly higher mean vaccine attitude scale score than mothers (*p* = 0.007).

Regarding education level, 25% of those with a primary/secondary education, 7.1% with a high school education, and 9% with a university education were convinced. A significant difference was observed among those with a primary/secondary education compared to other groups (*p* = 0.02).

Occupational groups also showed a significant difference in vaccine attitude scale scores (*p* < 0.001), primarily driven by lower scores among homemakers compared to other groups.

Regarding the number of children, 19% of those with one child, 4.7% with two children, and 9.9% with three or more children were convinced. The difference between those with one child and those with two children was significant (*p* = 0.024).

Participants with children under the age of one had a higher rate of persuasion (17.6%) compared to those without children under one (7.6%) (*p* = 0.034).

Regarding the perception of vaccine necessity, 21.6% of those who considered vaccination necessary, 7.6% of those who did not, and 7.1% of those who were undecided were convinced. A significant difference in persuasion was observed between those who considered vaccination necessary and the other two groups (*p* = 0.031).

Participants who had experienced problems with vaccines in the past had a lower rate of persuasion (4.5%) compared to those who had no problems (13.6%) (*p* = 0.015).

Regarding the belief in vaccine side effects, 7.1% of those who believed in side effects, 44.4% of those who did not, and 6.7% of those who were unsure were convinced. A significant difference was observed between these groups, primarily driven by the difference between those who believed they had no side effects and the other two groups (*p* < 0.001).

There was also a significant difference in vaccine attitude scale scores based on belief in side effects, with higher scores among those who believed in side effects than those who did not (*p* = 0.044).

Participants who had received some vaccinations had a higher rate of persuasion (11.6%) compared to those who had not received any vaccinations (2.6%) (*p* = 0.02) (Table 3).

The VAS score (*p* < 0.001) and age (*p* = 0.008) of those who were persuaded were found to be significantly lower than those who were not convinced (Table 4).

A significant positive correlation was observed between VAS and age, and it was determined that the VAS score increased as the age increased (r = 0.222; *p* < 0.001) (Figure 4).

## 4. Discussion

According to various studies, parents who express vaccine hesitancy or rejection due to the side effects of vaccines report a range of concerns. Some parents believe that vaccines can cause severe illness or pose a risk to their child’s health [13]. Others express concerns about the safety and potential side effects of vaccines, including seizures, myocarditis, stroke, and multisystem inflammatory syndrome in children (MIS-C) [14,15]. Additionally, some parents report concerns about the usefulness of vaccines and a lack of knowledge about recommended vaccinations [16,17]. In the present study, when parents who were hesitant about the vaccine were asked about the harms of the vaccine, 53% of the parents said that they did not know the harms of the vaccine, while the others gave different answers. After families are informed about the necessity of vaccines, they should be informed about their potential harm.

Based on the references provided, parents who express vaccine hesitancy or rejection due to the side effects of vaccines report a range of concerns. Some parents express concerns about the serious side effects of both routine childhood and influenza vaccines [18]. Others express concerns about the side effects of COVID-19 vaccines, including unknown effects and potential serious side effects [13]. In the present study, when parents who refused vaccination were asked about the vaccine’s side effects, the parents were most frequently concerned about side effects such as pain, fever, local redness, and infertility, respectively. The most frequently reported side effects are common and relatively simple, such as fever and pain. Health professionals should inform families that these side effects can be treated with simple medication if necessary and that this is not a situation to be feared.

Based on the references provided, the reasons for parents with childhood vaccine hesitancy or refusal not to get vaccinated include:-Safety concerns: Some parents express concerns about the safety of vaccines and their potential side effects [14].-Lack of need: Some parents may not see the need for vaccines or believe that their child is not at risk for vaccine-preventable diseases [19].-Negative media information: Exposure to negative information about vaccines through social media or other sources can contribute to vaccine hesitancy [20].-Fear of adverse events: Some parents may be hesitant to vaccinate their child due to the fear of adverse events following immunization (AEFIs) [21].-Time constraints: Parents may have difficulty finding time to take their child for vaccinations due to work schedules or other commitments [13].-Special physical conditions: Some parents may believe that their child’s physical condition is not suitable for vaccination [13].-Beliefs about risk: Some parents may believe that their child is not at risk for vaccine-preventable diseases or that the risk of severe illness from the disease is low [13].-Medical mistrust: Some parents may have a general mistrust of medical establishments or healthcare providers [22].-Cultural or religious beliefs: Some parents may have cultural or religious beliefs that conflict with vaccination [23].

In the present study, when parents who were hesitant about the vaccine were asked about the reasons, the most common reason was the vaccine’s side effects and the fact that the vaccine was not considered necessary, and the least common reason was religious reasons.

In 2019, a two-center study conducted in Turkey showed that vaccine refusal rates decreased as the education level of the mother and father increased. The same study found an inverse correlation between income level and vaccine refusal rate [10]. In our study, the persuasion rate of families with primary school graduates was significantly higher than the others. In another study conducted in Turkey in 2022, no correlation was found between educational status, income, and vaccination attitude/hesitancy [20]. The literature has drawn different conclusions on how education and income affect vaccination behavior and refusal [24].

In studies conducted in Turkey and worldwide, different determinations were made about vaccination attitudes according to gender. A study conducted in 2022 with nursing students in Turkey showed that males had a more negative vaccine attitude [25]. In a study conducted in 2022 in Turkey, no difference was observed in vaccination attitudes according to gender [24]. In our study, fathers were more convinced to vaccinate their children than mothers. If more studies on this subject are conducted and it is determined that fathers can be convinced more, information and persuasion programs that will appeal more to fathers about vaccine refusal can be implemented.

Comparing our study’s findings with those of other studies reveals some similarities and differences in the relationship between the age of parents and vaccine acceptance. In a study conducted by Soysal and Akdur, the frequency of factors related to vaccine hesitancy and refusal among parents of children under five was examined [23]. They found vaccine hesitancy was higher among parents over 42 years of age. This finding contrasts with our study, which suggests that as parents’ ages decrease, the rate of being convinced about the vaccine increases. It is important to note that this study focused on parents of children under five, which may explain the difference in findings. A study by Lu et al. explored parental attitudes toward vaccination against COVID-19 in China. They found that parents’ willingness to accept vaccinations for their children increased with the children’s age. It indicates that parental attitudes towards vaccination may vary depending on the age of their children [26]. In the study by Loncarevic et al., parental knowledge and attitudes toward the MMR vaccine in Serbia were examined. While their study did not specifically focus on parents’ age, they found that getting information on vaccination from a pediatrician was associated with an increased probability of vaccinating a child. This suggests that healthcare providers, including pediatricians, play a crucial role in shaping parental attitudes toward vaccination [27]. Comparing the findings from our study with other studies in the literature reveals some similarities and differences. While some studies did not directly examine parents’ ages, they provided insights into parental vaccine hesitancy and acceptance. The relationship between parents’ age and vaccine acceptance may vary depending on the specific context and population studied. Factors such as the type of vaccine, the age of the child, and the role of healthcare providers can influence parental attitudes toward vaccination.

The literature provides some insights into the vaccination attitudes of homemakers, although the specific focus on homemakers may be limited. In a study by Abedin et al., which examined the willingness to vaccinate against COVID-19 among Bangladeshi adults, it was reported that homemakers had a high level of resistance to being vaccinated against COVID-19 [28]. This finding aligns with our study, which suggests that homemakers have higher vaccination hesitancy and refusal. It indicates that homemakers may exhibit more reluctance toward vaccination than other groups. A systematic review by Tamara et al. investigated the refusal of the COVID-19 vaccination and its associated factors. The review reported that females, including homemakers, lacked literacy regarding COVID-19 vaccination compared to males [29]. This finding suggests that homemakers may have less knowledge and information about COVID-19 vaccination, which could contribute to their vaccination hesitancy. Another study by Bukhari et al. (2020) examined the effectiveness of health education messages in improving tetanus health literacy among women of childbearing age. While the study did not specifically focus on homemakers, it provides insights into the impact of health education messages on women’s health literacy. The findings indicated that the health education message was more effective for students and employees than for homemakers [30]. This suggests that targeted health education interventions may be necessary to improve vaccination attitudes and knowledge among homemakers. In parallel to our research, the available literature provides some insights into the vaccination attitudes of homemakers. While some studies focused on homemakers, others indirectly addressed their attitudes through related topics. We suggest that homemakers may exhibit higher vaccination hesitancy and resistance, potentially due to limited access to information, engagement with lifestyle influencers, and lower health literacy. However, further research explicitly targeting the vaccination attitudes of homemakers is needed to gain a more comprehensive understanding of this population’s perspectives.

The finding from our study, which suggests that parents with children under one year of age are more likely to be persuaded for childhood vaccination, aligns with some findings in the literature. A systematic review by Schmid et al. identified barriers to influenza vaccination uptake or intention. While the study focused on influenza vaccination, it provides insights into parental decision-making for childhood vaccination. The review found that parental decisions for high-risk children were under-represented in the literature, indicating a gap in understanding this specific population [31]. This suggests that limited research may examine the vaccination decisions of parents with children under one year of age. Further research is needed to better understand the factors influencing parental decision-making for childhood vaccination, particularly for children under one year of age.

The finding from our study, which suggests that people who think that vaccination is necessary but are hesitant about it are more likely to be convinced than people who think that vaccination is not necessary, is supported by some findings in the literature. A study by Brewer et al. discusses the concept of “reluctant but persuadable” individuals who hold ambivalent or unfavorable beliefs about a vaccine but can be convinced. The study highlights the challenges and limited effectiveness of persuasive communications in changing the beliefs and attitudes of these individuals [32]. This aligns with the finding that hesitant individuals who recognize the necessity of vaccination may be more open to persuasion. In a randomized controlled trial by Freeman et al., different types of written information about COVID-19 vaccinations were tested to increase vaccine acceptance. The study found that providing information about the pandemic’s collective benefit, personal benefit, seriousness, and safety concerns increased vaccine acceptance. This suggests that addressing the reasons for hesitancy and providing relevant information can effectively convince hesitant individuals [33]. The literature supports the idea that individuals who acknowledge the necessity of vaccination but are hesitant may be more open to persuasion.

The finding from our study, which suggests that those who had previously experienced problems during vaccination had lower rates of persuasion, is supported by some findings in the literature. A study by Baldwin et al. focused on promoting adolescent HPV vaccination among parents attending safety-net clinics. The study found that many parents remained undecided or ambivalent about the vaccine, even with a provider’s recommendation [34]. This indicates that previous experiences or concerns may contribute to vaccine hesitancy and lower persuasion rates. In another study by Argyris et al., the authors used machine learning to compare pro- and anti-vaccine discourses on social media. The study aimed to understand the attempts made by advocates to influence the public’s acceptance or rejection of immunization [35]. This suggests that previous negative experiences or concerns about vaccines may shape individuals’ attitudes and beliefs, leading to lower persuasion rates. Zhou et al. conducted a multicenter survey among Chinese medical students to examine HPV vaccine hesitancy. The study found that convenience was the primary factor in vaccine hesitancy, and many students relied on a single source for vaccine information [36]. This suggests that previous negative experiences or concerns may contribute to vaccine hesitancy and lower persuasion rates among medical students. The literature supports the finding that previous negative experiences during vaccination can lead to lower persuasion rates, paralleling our study. Understanding these factors and addressing them through targeted interventions and communication strategies can help improve vaccine acceptance and address concerns related to previous negative experiences.

The finding from our study, which suggests that parents who were partially vaccinated were more likely to be convinced than those who did not vaccinate at all, is supported by some findings in the literature. Ellithorpe et al. conducted a study in the United States to compare vaccine acceptance behavior across different groups, including those who were fully accepting of the recommended vaccination schedule, those who were accepted but on a delayed schedule, those who were only partially vaccinated, and those who did not vaccinate at all. The study found apparent differences between the vaccination behavior groups, with financial and insurance-related barriers hindering complete vaccination [37]. This suggests that partially vaccinated individuals may be more open to accepting further vaccination compared to those who did not vaccinate at all, paralleling our study. An et al. conducted a study in China and found that parental vaccine hesitancy led to a refusal to use COVID-19 and influenza vaccines. The study also found that parents were partially hesitant to vaccinate their children with COVID-19 [34]. This suggests that individuals who have had some level of vaccination may be more open to accepting further vaccination than those who have not been vaccinated, which is also similar to our study. Raof conducted a study in Iraq and found that a higher percentage of children were fully vaccinated among parents with lower vaccine hesitancy scores than among parents with higher scores [38]. This indicates that parents who were partially hesitant about vaccination were more likely to have their children fully vaccinated than more hesitant parents. While the literature provides some support for the finding that parents who were partially vaccinated are more likely to be convinced than those who did not vaccinate at all.

Based on the literature provided, the post-training vaccine acceptance rates vary across different studies and populations. Hallas et al. investigated targeting vaccine-hesitant prenatal women and mothers of newborns and found that 82% of prenatal vaccine-hesitant women had full prenatal vaccination coverage after receiving the intervention [39]. Sitaresmi et al. (2020) investigated HPV vaccination in Indonesia and reported improved parental awareness, knowledge, perception, and acceptability of HPV vaccination after the intervention [40]. Nesiama et al. (2022), who conducted a survey among personnel associated with Wright–Patterson Air Force Base, found that 64% of participants remained hesitant following a series of seminars [41]. Comparing these findings with the 9.3% post-training vaccine acceptance rate reported in our research, it falls within the range of vaccine acceptance rates reported in the literature. However, it is essential to note that the literature includes studies on various vaccines, populations, and interventions, which may contribute to the variability in acceptance rates. Additionally, the specific details of the training provided in our research and the characteristics of the study population also influence the observed acceptance rate.

Based on the references provided, there is a relationship between parents with childhood vaccine hesitancy/refusal, their persuasion to get vaccinated, and the vaccine attitude scale. Studies have identified a spectrum of parent attitudes or “positions” on childhood vaccination, with estimates of the proportion of each group based on population studies [42]. Some studies have found no evidence of an association between parental vaccine hesitancy and adolescent vaccine uptake [43]. Parents who were hesitant about childhood vaccinations had lower positive attitudes toward the COVID-19 vaccine than parents who were not hesitant [20]. Perceived financial well-being had significant and negative associations with parents’ attitudes toward COVID-19 vaccines and child vulnerability [44]. Mistrust of the medical profession and higher disgust sensitivity predict parental vaccine hesitancy [45]. Overall, understanding the relationship between parents with childhood vaccine hesitancy/refusal, their persuasion to get vaccinated, and the vaccine attitude scale is crucial in developing strategies to improve childhood vaccination rates and promote public health. In our study, parallel to the literature, a correlation was shown between the vaccination attitude scale and persuasion.

Since our study was conducted in a narrow geographical area, sociocultural factors may have come to the fore. As the study area is relatively rural, results may differ in urban areas. Since the home visits were conducted during working hours, only one of the parents could be interviewed, and the interviewed parent was generally the mother since the working population in Turkish society is primarily male. The study’s results may not be generalizable since not all families who refused vaccination could be interviewed, and only vaccine refusals in one province were examined.

## 5. Conclusions

Our study aims to determine the vaccine attitudes of families who refuse vaccination as well as plan for the modification of decisions regarding vaccine refusal among the families interviewed. The intention is to provide guidance for future vaccine persuasion efforts by identifying more persuadable groups. In our study, the persuasion rates of families informed by face-to-face interviews with the vaccine attitude scale were followed by six-month records. The rates were determined according to these records. Our research has shown that face-to-face meetings and education are useful options for convincing people who are hesitant about vaccination.

## Figures and Tables

**Figure 1 vaccines-11-01285-f001:**
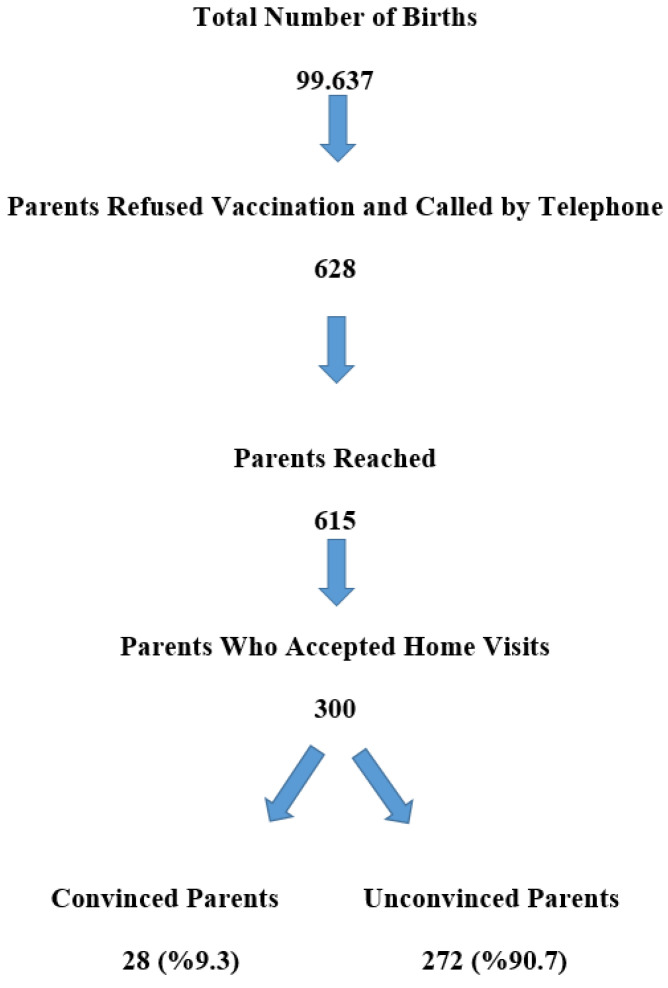
Study Flow Diagram.

**Figure 2 vaccines-11-01285-f002:**
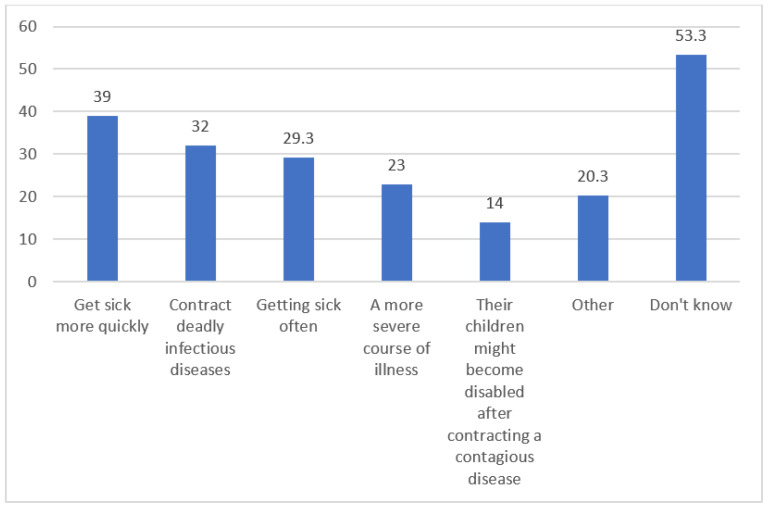
Opinions of the participants about the harms of the vaccine.

**Figure 3 vaccines-11-01285-f003:**
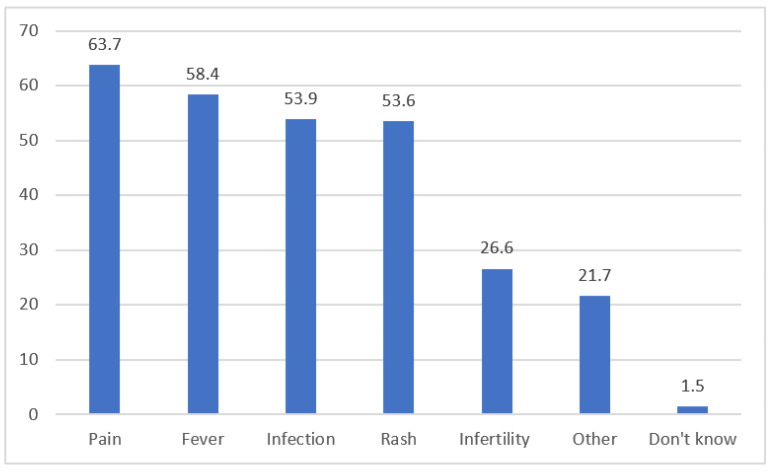
Opinions of the participants about the side effects of the vaccine.

**Figure 4 vaccines-11-01285-f004:**
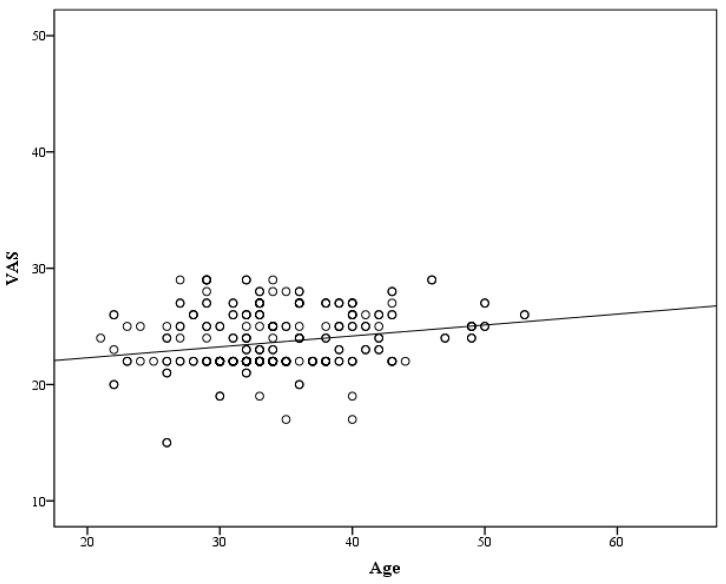
Correlation of VAS with age.

**Table 1 vaccines-11-01285-t001:** All characteristics of the participants in the study.

		Number	%
Age, Mean ± SD		34.2 ± 5.8	
Interviewed person	Father	52	17.3
	Mother	248	82.7
Educational status	Primary/secondary school	24	8.0
	High School	154	51.3
	University	122	40.7
Profession	Homemaker	180	60.0
	Teacher	36	12.0
	Government employee	43	14.3
	Healthcare profession	14	4.7
	Artisan	16	5.3
	Other	11	3.7
Number of Children	1	42	14.0
	2	107	35.7
	3 and above	151	50.3
Presence of children under one-year-old	Yes	51	17.0
	No	249	83.0
Opinion on necessity of the vaccine	Yes	37	12.3
	No	249	83.0
	Indecisive	14	4.7
Having a child vaccinated previously.	Yes	264	88.0
	No	36	12.0
Having problems with vaccination before	Yes	110	41.7
	No	154	58.3
Status of getting the vaccines recommended by the Ministry	Has gotten some vaccines	224	74.7
	Has not gotten any vaccines	76	25.3

**Table 2 vaccines-11-01285-t002:** Reasons for not getting vaccinated by the study participants *.

	Number	%
I do not think vaccines are safe/I am worried about the side effects	297	99.0
I think it is not necessary	215	71.7
I do not think vaccines are effective in protecting from diseases	183	61.0
Chemical substances in vaccines are harmful to human health.	151	50.3
Someone else said that their child had a bad reaction after vaccination.	133	44.3
Negative things about the vaccine I read or heard from the media	110	36.7
Bad experiences with healthcare professionals or healthcare provider	71	23.7
Religious reasons	24	8.0

* some participants have more than one reason for not being vaccinated.

**Table 3 vaccines-11-01285-t003:** Comparison of the persuasion status and vaccine attitude scale scores of the participants according to various parameters.

		Those Who Are Convinced		*p* *	VAS	*p* **
		Number	%		Mean ± SD	
Interviewed person	Father	9	17.3	0.038	**24.3 ± 3.5**	0.007
	Mother	19	7.7		23.5 ± 2.1	
Educational status	Primary/secondary school	6	**25.0 ^a^**	0.02	23.5 ± 2.5	0.258
	High School	11	7.1 ^b^		23.5 ± 2.1	
	University	11	9.0 ^b^		23.8 ± 2.8	
Profession	Homemaker	19	10.6	0.745	**23.0 ± 2.0 ^a^**	<0.001
	Teacher	3	8.3		25.0 ± 2.6 ^b^	
	Government employee	4	9.3		24.2 ± 2.5 ^b^	
	Healthcare profession	2	14.3		24.2 ± 4.5 ^b^	
	Artisan	0	0.0		24.4 ± 2.1 ^b^	
	Other	0	0.0		26.1 ± 1.8 ^b^	
Number of Children	1	8	**19.0 ^a^**	0.024	23.5 ± 1.7	0.480
	2	5	4.7 ^b^		23.9 ± 2.7	
	3 and above	15	9.9 ^ab^		23.5 ± 2.4	
Presence of children under one-year-old	Yes	9	**17.6**	0.034	24.0 ± 3.1	0.280
	No	19	7.6		23.6 ± 2.3	
Perception on necessity of the vaccine	Yes	8	**21.6 ^a^**	0.031	23.4 ± 3.0	0.128
	No	19	7.6 ^b^		23.6 ± 2.3	
	Indecisive	1	7.1 ^b^		24.5 ± 2.8	
Having a child vaccinated previously.	Yes	26	9.8	0.551	23.7 ± 2.5	0.202
	No	2	5.6		23.2 ± 1.7	
Having problems with vaccination before	Yes	5	4.5	0.015	23.7 ± 2.4	0.937
	No	21	13.6		23.7 ± 2.7	
Is there any side effect?	Yes	19	7.1 ^a^	<0.001	23.8 ± 2.3 ^a^	0.044
	No	8	**44.4 ^b^**		**21.8 ± 3.4 ^b^**	
	I do not know	1	6.7 ^a^		22.9 ± 2.2 ^ab^	
Status of getting the vaccines recommended by the Ministry	Has gotten some vaccines	26	11.6	0.02	23.7 ± 2.5	0.320
	Has not gotten any vaccines	2	2.6		23.5 ± 2.4	

* Chi-square analysis, ** Mann–Whitney U test in pairwise comparisons, Kruskal–Wallis analysis in more than two comparisons. Row percentage is used. ^a,b^ It is the parameter caused to statistical differences.

**Table 4 vaccines-11-01285-t004:** Comparison of VAS and age according to persuasion status.

	Those Who Are Convinced	Those Who Are Not Convinced	*p* *
	Mean ± SD	Mean ± SD	
VAS	20.7 ± 2.2	23.9 ± 2.3	<0.001
Age	31.4 ± 5.3	34.5 ± 5.8	0.008

* Mann-Whitney U test was applied.

## Data Availability

Data will be available upon request from the corresponding author.

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
