# Peer review of "Childhood Vaccine Attitude and Refusal among Turkish Parents"

_vaccines, 2023, doi:10.3390/vaccines11081285_

Round 1

Reviewer 1 Report

MDPI

Vaccines

Review Letter to Authors

Manuscript Title:          Childhood vaccine hesitancy and refusal among Turkish parents

Manuscript ID:             Not Provided

Date:                              Sunday, June 18, 2023

_________________________________________________________

The authors aimed to understand and resolve anti-vaccine hesitancy by examining the factors associated with vaccine hesitancy and exploring potential strategies to improve childhood vaccination rates. Between 2014 and 2021, a total of 628 families refused vaccination in Adıyaman province, located in the Southeastern Anatolia region of Turkey. A total of 300 families were visited and during the visits, the families were administered a questionnaire to determine the reasons for vaccine rejection and their opinions on the matter. While providing general information about the vaccine, parents were encouraged to reconsider their decision, and at the end, parents completed the questionnaire. Participants who had received some vaccinations had a higher rate of persuasion (11.6%) compared to those who had not received any vaccinations (2.6%) (p=0.02).

I found this topic interesting and enjoyed reading this manuscript. It is in the spirt of strengthening the manuscript that I provide the following recommendations and pose two questions:  

Your Title Is:

Childhood vaccine hesitancy and refusal among Turkish parents

Change to:

Childhood Vaccine Hesitancy and Refusal among Turkish Parents

On Page 1, you wrote:

A total of 300 families accepted visits were visited.

Change to:

A total of 300 families accepted visits and were visited.

OR

A total of 300 families were visited.

On Page 1, you wrote:

This descriptive study was conducted in the Adiyaman province, located in the Southeastern Anatolia region of Turkey.

 ·        Provide some information regarding the prevalence of vaccines in this region.

On Page 2, you wrote:

Initially, families were contacted by phone and informed that they would be visited at their residences.

·        Were all families visited in their residences? Did some families choose a different location than their residence? If so, where? This must be clear.

On Page 2, you wrote:

A total of 300 families accepted visits were visited.

Change to:

A total of 300 families accepted visits and were visited.

OR

A total of 300 families were visited.

ADDITIONAL QUESTION: Who visited these parents? Did one or more of the authors of this manuscript visit these parents? Please be clear regarding who did these visits, the number of visits that were made to these parents, as well as the length of time that these visits occurred.

On Page 2, you wrote:

While providing general information about the vaccine, parents were encouraged to reconsider their decision, at the end parents completed the questionnaire.

·        Exactly how were parents encouraged to reconsider their decision? What was said to these parents? What information was shared with these parents?

On Page 5, you wrote:

Among the participants in the study, 9.3% were convinced about the vaccine.

·        This is not clear. Reword for clarity. Are you saying that 9.3% of participants were convinced that they should get the vaccine OR should not get the vaccine?

THEORETICAL FRAMEWORK. Your work would be stronger if it were based on a theoretical framework.

DISCUSSION

·        Limitations of the Current Study. All studies have limitations. What were some limitations of your study?

·        Directions for Future Research. In what ways can scholars build on and/or extend the findings of your study?

·        Practical Implications. In what ways can the findings of your study inform the work of medical professionals (i.e., clinicians)?

THERE ARE SEVERAL APA ISSUES WITHIN THE PAPER AND ON THE REFERENCE PAGE. THE AUTHOR MUST CORRECT THESE ERRORS.

According to the 7th edition of APA:

1.       You should alphabetize in-text citations

2.       You should italicize journal titles.

3.       You should italicize the volume number.

4.       You should not italicize the issue number.

5.       You should keep the volume number AND issue number together.  

6.       You should italicize book titles.

7.       You should capitalize journal titles.

8.       You should add a comma after the initial of an author

9.       You should use the & on your Reference page and not “and” [The only time that you use “and” on your Reference page is if “and” in the actual title of the work being cited]

10.     You should not use “Retrieved from”

11.     You should provide beginning and ending page numbers.

12.     When citing 3 or more authors, you should use the surname of the first author followed by et al and the year [This is the case for in-text citations and citations within a sentence].

13.     You should not capitalize every word in your title. Only capitalize the proper nouns.

FINAL NOTE: Please check to make sure that all citations in the paper are also on the Reference pages.

Author Response

Dear Reviewer

Many thanks for your contribution to our work. Revisions I made in the study are listed below:

Since the vaccine attitude scale was used, the study's title was changed to “Childhood Vaccine Attitude and Refusal among Turkish Parents”.

The mentioned typos have been corrected.

In the material method section, who visited the families, where the interviews were held, and how long they lasted were added. Statements about those who were persuaded were edited.

The Study Flow Diagram has been added, and the study design has been explained in more detail.

The Discussion section has been expanded. A limitation has been added. A paragraph has been added for direction for future research. A paragraph has been added for practical implications.

The conclusion part has been revised and edited.

The revisions mentioned in the references section could not be made because this section was written according to the journal's rules.

English of the study has been revised.

Osman Küçükkelepçe, MD, PhD.

Reviewer 2 Report

Thank you for the invitation to review this manuscript. 

Introduction: The introduction section does not provide rationale of this study. The authors need to describe the number of studies already conducted in Turkey on the same topic. What literature gap is actually covered by this study? What is the current situation of childhood vaccination in the country and why the authors inclined to conduct the study in relation to that situation. The introduction section is very brief and generic rather than addressing the problem area of this study.

Methods: The methods section is also very brief. The authors need to address following headings; ethics, study design and location, study population (inc. and exc. criteria), Sample size estimation, Study instrument (questionnaire) - including components and its validation, reliability and translation, Data collection and sampling method, outcome measures or operational definitions used in this study and statistics.

It is not clear that the parents were either subjected to interviews or a self-administered questionnaire methods were used.

The authors used the word persuasion, but did not define it in the method section. What do they mean about persuasion and how did they measure it?

How the authors classified the participants as convinced? The definitions of convinced and not convinced should be described in the methods section.

Please provide the study flow diagram in the methods section.

Please mention the data collection time too

Results

I am not sure that how the attitude towards the vaccine can predict the vaccine hesitancy. There are two terms here, one is vaccine attitude and other is vaccine hesitancy. The authors have claimed that negative vaccine attitude is considered as vaccine hesitancy. However, I believe that the authors need some references of studies where attitude is correlated with the vaccine hesitancy.

I am also not sure why the authors did not use the VH scale. There are numerous well-established scales to measure the VH but the authors preferred to consider the attitude scales. 

Discussion

The discussion section is not well written. The authors need to compare the results with other Turkish studies and then the studies conducted elsewhere. Please provide the limitations of the study. The sample size is low, and the participation of the father is also less to make any firm conclusion. The authors have claimed differences between father and mother regarding vaccine attitude, but there is a need to consider the low sample size of fathers in this study. The authors should also compare these results with other studies, and explain the possible reasons of positive attitude among fathers as compared to mothers. The same approach should be used for other major results of this study. 

Conclusions

The conclusion is very generic and only provides general information rather than major findings originated from the current study. The authors are encouraged the write the conclusions again with a start of major results, followed by directions for healthcare professionals, policies and future research.

A little bit improvement needed.

Author Response

Dear Reviewer

Many thanks for your contribution to our work. Revisions I made in the study are listed below:

Since the vaccine attitude scale was used, the study's title was changed to “Childhood Vaccine Attitude and Refusal among Turkish Parents”.

The introduction expanded by mentioning the studies and current situation in Turkey.

The material method section is structured.

The Study Flow Diagram has been added, and the study design has been explained in more detail.

The application method of the questionnaire was explained.

Statements indicating parents are convinced been added.

The control method of those who were persuaded was explained.

The time to perform visits has been added.

The Vaccine Attitude Scale was used in the study, and its relationship with vaccination hesitancy was cited in reference number 12. This is mentioned in the material method section.

The discussion section has been expanded in line with the suggestions. Some repetitive parts have been deleted.

Limitation has been added.

The conclusion part has been revised and edited.

English of the study has been revised.

Osman Küçükkelepçe, MD, PhD.
